# Chlorophyll Fluorescence Imaging for Environmental Stress Diagnosis in Crops

**DOI:** 10.3390/s24051442

**Published:** 2024-02-23

**Authors:** Beomjin Park, Seunghwan Wi, Hwanjo Chung, Hoonsoo Lee

**Affiliations:** 1Department of Biosystems Engineering, College of Agriculture, Life & Environment Science, Chungbuk National University, 1 Chungdae-ro, Seowon-gu, Cheongju-si 28644, Republic of Korea; 2Vegetable Research Division, National Institute of Horticultural & Herbal Science, Wanju 55365, Republic of Korea

**Keywords:** nondestructive evaluation, garlic, chlorophyll fluorescence, near ultra-violet, photon excitation, environmental stress, multispectral imaging, partial least squares discriminant analysis (PLS-DA), classification

## Abstract

The field of plant phenotype is used to analyze the shape and physiological characteristics of crops in multiple dimensions. Imaging, using non-destructive optical characteristics of plants, analyzes growth characteristics through spectral data. Among these, fluorescence imaging technology is a method of evaluating the physiological characteristics of crops by inducing plant excitation using a specific light source. Through this, we investigate how fluorescence imaging responds sensitively to environmental stress in garlic and can provide important information on future stress management. In this study, near UV LED (405 nm) was used to induce the fluorescence phenomenon of garlic, and fluorescence images were obtained to classify and evaluate crops exposed to abiotic environmental stress. Physiological characteristics related to environmental stress were developed from fluorescence sample images using the Chlorophyll ratio method, and classification performance was evaluated by developing a classification model based on partial least squares discrimination analysis from the image spectrum for stress identification. The environmental stress classification performance identified from the Chlorophyll ratio was 14.9% in F673/F717, 25.6% in F685/F730, and 0.209% in F690/F735. The spectrum-developed PLS-DA showed classification accuracy of 39.6%, 56.2% and 70.7% in Smoothing, MSV, and SNV, respectively. Spectrum pretreatment-based PLS-DA showed higher discrimination performance than the existing image-based Chlorophyll ratio.

## 1. Introduction

Plants growing in nature are predominantly exposed to a host of stresses, primarily due to abiotic factors such as nutrition, temperature, moisture, etc., and biological ones including microorganisms, parasites, insects, among others [1,2,3]. These stresses impede plant growth and productivity and may induce alterations at the physiological, biochemical, and molecular levels of plants. In particular, abiotic stresses like high heat and moisture stress exert a more significant impact on crops, influencing the emergence and spread of pathogens, insects, and weeds. Nonetheless, complex stress factors do not invariably exert a negative effect on crops, and the capacity to adapt and recover differs based on the interaction among each stress factor [4,5]. Therefore, plants exhibit a unique response to stressors, and understanding the complexity of these interactions is crucial for effectively dealing with plant stress. Additionally, although the combination of high heat and moisture stress has been found to modify physiological and molecular processes such as photosynthesis and transcriptome expression, knowledge about traits and growth characteristics is not well-established [6,7,8].

Plant Phenotyping is a field of biology that analyzes phenotypic image information such as the physiological state, shape, and biochemical characteristics of crops for data analysis optimized for crop cultivation. This field comprehensively interprets physical, morphological, physiological, and biochemical characteristics that appear in cells, tissues, institutions, and living things and is a convergence technology using engineering, environmental control, and optical technology [9]. Through this, it is being used as a high-efficiency technology to measure and evaluate automated plant phenotypes by capturing images related to plant growth and combining bio-information technology. These expressive technologies are gradually expanding their meaning and scope of application [10,11,12]. Plants may have qualitative or quantitative physiological characteristics for responses that may be caused by abiotic (environmental factors such as nutrition, temperature, humidity, and soil moisture content) and biotic (microorganisms, parasites, insects) during growth. It is used to allow appropriate action to be taken from visual assessment of the severity of disease and stress through quantitative measured phenotypes [13].

Among them, quantitative phenotypic monitoring using video technology can offer various benefits, but it requires accurate data processing and integration suitable for the environment [6]. Modern high-resolution imaging technology approaches enable the visualization of multidimensional and multi-data on the spectrum, and these techniques are mainly utilized to measure the parameters of plant-related bio-information or the complex properties of stress quantification [14]. Plants respond to subtle changes in the light environment, and when light passes through various layers of tissue, their properties are altered according to the optical properties of each layer [15]. 

The majority of photosynthetic organisms utilize chlorophyll in the process of collecting light associated with the photometer, and plant parameters observe metabolic information associated with it to determine physiological variability [16,17]. Light-induced light energy is distributed through three processes: photosynthesis, heat release, and the fluorescence emission of chlorophyll [18]. The absorption of photons promotes the electrons in the chlorophyll molecule to an excited state, while the fluorescent strongman is emitted again while the molecule returns to the ground state. Chlorophyll fluorescence techniques provide insight into activation energy utilization through photosystem 2 as well as indirect utilization through various complexes within the thylakoid membrane. These techniques, known as Fluorescence Induction (FI), contribute greatly to clarifying plant photosynthesis and important physiological metabolic mechanisms in combination with other spectroscopic and biochemical methods [19]. Chlorophyll fluorescence technology is used as an important tool to understand energy transfer and transformation and optical processes in photosynthesis through how chlorophyll molecules respond when exposed to light [20]. Fluorescence yields are variable as a result of photochemical and non-photochemical processes, and it is possible that inhibition of metabolic reactions not involved in photosynthesis feeds back the rate of synthesis of metabolic intermediates and consequently inhibits photosynthesis and fluorescence emission functions [21,22,23]. Chlorophyll fluorescence imaging (CFI) is an imaging of fluorescence-sensitive signals caused by metabolic disturbances between photosynthetic respiratory processes and acquires spatial variability information of fluorescence patterns to measure changes in specific spectral regions for disease and stress caused by abiotic and biological factors [24]. Fluorescence imaging is used to monitor non-photosynthesis-related pathogens and environmental stresses by inducing fluorescence re-emission signals from plants by illuminating samples with visible or UV light via pulsed lasers or LEDs [25]. Fluorescence imaging is useful for the early identification of factors affecting plant health and productivity through the analysis of relationships with phenotypic data, by detecting photosynthetic-associated pathogens and environmental stresses through fluorescence re-emission signals in plants [26].

Chlorophyll fluorescence imaging tends to require LEDs for uniform active illumination and fully saturated pulses across the entire imaging area. Among them, UV LEDs have been known as a good way to investigate the environmental response to compound concentrations and properties from plant epidermal screening [27]. UV light is divided into UV-C (<280 nm), UV-B (280–320 nm), and UV-A (320–380 nm). Bushman and Richtenhaler used a UV radiation imaging system (fluorescence imaging) for physiological measurements and early stress detection by screening the fluorescence signals of leaves, and they secured high statistical reliability at a ratio according to specific fluorescence sensitivity [28]. Guidi et al. [29] applied fluorescence imaging to investigate the photosynthetic activity stress of abiotic plants and ozone-induced photosynthetic disturbances induced by pathogens to confirm the reaction of the light system affecting the entire surface, showing that the heterogeneity of the fluorescence yield was clearly observed. Hisao et al. [30] attempted to non-destructively evaluate moisture stress by inducing the release of fluorescent chlorophyll in cabbage seedlings, using blue LEDs in the darkroom. In the dynamic fluorescence image index system equipped with a multispectral imaging system, a quantitative model was developed to predict the moisture stress state of cabbage seedlings in 720 nm channels as a result of analyzing the physiological state by introducing the ratio of the maximum value of the fluorescence level. One study by He et al. [31] provided the applicability of a fluorescence multispectral reflection imaging platform to detect viral infection in citrus leaves with a 380 nm UV LED. However, these UV regions still have wavelengths that can cause crop damage, and fluorescence-activated induced light sources are needed to replace them.

Thereto, ref. [32] effectively detected the effect of nanoparticle content on soybeans on plants from UV-B light sources, using a fluorescence activation technique using a 405 nm light source. Ref. [33] succeeded in evaluating the amount of physiological stress caused by damage to photometer 2 of corn. In addition, ref. [34] successfully detected damage to *Jatropha curcas* L., due to a lack of water after applying a fluorescence ratio using a blue lead with a center wavelength of 405 nm. Ref. [35] established a classification model of pathogen infection and non-infected leaves through 405 nm LED-based chlorophyll activity image analysis on potato leaves. Since the CFI system can capture fluorescence intensity value from each waveband, a multivariate calibration, such as the partial least squares (PLS) algorithm, typically assists the computation. It conceptually works by deflating the high dimensionality of the original matrix; therefore, a less complex model can be established. Ye Sun et al. [36] showed excellent performance by establishing a chlorophyll fluorescence imaging system to discriminate against diseases of cucumber plants, and they established a classification model between fluorescence parameters and stress. There are methods using PLS for the optimal separation of classes using spectra in the creation of classification models among various variables and for the technique approach of selecting variables in spectra [37]. PLS is known as the most useful method for infection and detection, using the ability of pattern recognition of spectra [38,39,40]. As well as regression tasks, a classification model, PLS-DA, is also applicable. Kim et al. [41] developed a spectral image classification model using PLS-DA to screen for the heat stress in cabbage early and showed high classification accuracy. Kasampalis et al. [42] developed a parameter classification model to induce spectral chlorophyll fluorescence and confirm freshness through a spectral camera, presenting reliable evaluation. The performance of biomass and classification models using the fluorescence spectrum has been proven, but ultraviolet rays still have effects on plants, resulting in damage and limitations. Moreover, fluorescence analysis studies on environmental stress for specific crops are insignificant. 

Accordingly, this study focuses on the development of the chlorophyll fluorescence ratio system of crops, explores whether Near UV radiation outside the ultraviolet region affects the stress classification and performance of crops, and combines the spectrum-based environmental stress phenotype classification model with the chemical analysis method of PLS discriminant analysis (PLS-DA) to create an optimization model and applies it to classify and interpret the stress of crops. 

## 2. Materials and Methods

### 2.1. Garlic Cultivation

The study area was located at the National Institute of Horticulture and Herbal Science Republic of Korea (see Figure 1). After formalization, garlic was grown outside for 210 days. A composite soil was used at a ratio of 5:5 of red clay and sands. The garlic used grows in warm climates regions, specifically “Namdo”, a variety that grows in warm climates. Garlic in the hematocrit was administered for seven days under conditions that were controlled in a high-temperature enclosed weather chamber (see Table 1). In order to create a temperature environment for outdoor field cultivation, Group 1 maintained 20 ℃ during the day and 8 °C during the evening. Group 2 experienced high-temperature stress of 28 °C during the day and 16 °C in the evening. Group 3 was set at 36 °C in the daytime and 24 °C in the evening. In this study, soil moisture levels were treated by applying moisture. To maintain normal soil conditions with a moisture potential of 0 kPa, and to treat soil conditions with a moisture potential of 30 kPa, fresh-water irrigation was applied to the treatment pots during the first week of each month.

### 2.2. Chrolophyll Fluorescense Spectral Imaging System

The chlorophyll fluorescence excitation phenomenon transition energy principle was used to non-destructively examine abiotic stress in crops. This refers to the fluorescence spectrum emitted according to the wavelength of light absorbed by chlorophyll molecules during the photosynthesis process. As mentioned, a chlorophyll fluorescence mode imaging system was used to scan canopy images of crops. The fluorescence modules used in the experiment are shown in Figure 2. The camera used in this study was measured by a visible-light camera sensor with a 25-channel acquisition capacity in the wavelength range of 603 nm to 870 nm and a 2048 × 1088-resolution visible/near-infrared imaging camera (OCI-D2000; Bayspec Inc., San Jose, CA, USA). Canopy images were captured using software (Bayspec Frabber ver 1.0.2, Bayspec Inc., San Jose, CA, USA), and the distance between garlic samples from the lens of the camera was adjusted to 20 cm for the smallest individual, and the camera exposure time was set to 100 ms. The light source used in this system was a customized Near Ultraviolet LED. Each LED CA3535(CUN0GF1A; Seoul Viosys Co., Ltd., Seoul, Republic of Korea) has a maximum output of 10 W, and 24 LEDs are used in one module. Heat dissipation is used to maintain a constant light source output by naturally radiating heat using spin cooling fins. Garlic was in a dark-adapted state for 12 h, and photographs were taken after light adaptation for 5 min under chlorophyll fluorescence activation.

### 2.3. Fluorescence Image Data Correction

Multispectral fluorescence images were subjected to photo-correcting processes to correct images of garlic leaves due to uneven intensity and noise of the camera’s light source during collection. Calibration was performed using the PTFE board (manufactured by Labsphere Co., Ltd., North Sutton, NH, USA). This process was performed using Equation (1) to modify the raw time multispectral image, which was performed on image ${\mathrm{MSI}}_{f}$ using the Bayspec Cube Creator 2100 (Bayspec Inc., San Jose, CA, USA). Subsequently, the region of interest (ROI) was chosen randomly from the leaves that had been exposed to environmental stress. Data acquisition, calibration, and extraction are shown in Figure 3.
(1)$${\mathrm{MSI}}_{f}=\frac{{\mathrm{MSI}}_{\mathrm{raw}}-{\mathrm{MSI}}_{d}}{{\mathrm{MSI}}_{w}-{\mathrm{MSI}}_{d}}$$
where ${MSI}_{raw}$ is the raw image of garlic leaves acquired from the camera system. ${MSI}_{w}$ is a fluorescence image obtained from a PTFE solid, and ${MSI}_{d}$ is a dark reference image. The dark reference image was acquired with a lens covered by a cap.

### 2.4. Chlorophyll Fluorescense Ratio

Chloroplasts, which play an essential role in the plant response to abiotic stress, have been proposed as phenotypic indicators of plant stress against environmental changes [43]. Among them, the chlorophyll fluorescence ratio analysis method is an analysis method that estimates the photosynthetic function mechanisms for plant stress as a ratio by measuring the biological response expressed in the chloroplasts in a spectrum [44]. Chlorophyll fluorescence can detect stress before visual symptoms occur at the user’s point in time and is ideal for early identification of individuals whose abiotic features are not expressed [45]. In addition, by visualizing the color of leaves, it is an efficient alternative for conveying stress information to users and for evaluating from media that cause physiological disorders. In this study, a Near UV LED CA3535 with a wavelength of 405 nm was used for fluorescence imaging and has a maximum output of 240 W. Heat dissipation was used to keep the light source output constant by natural heat dissipation using spin cooling fins. Garlic was in a dark adaptation state, and light was adapted for 1 min. The measured fluorescence spectrum parameters were calculated as the fluorescence ratio, and the equation is as follows: Equation F690/F735 has proven the usefulness of D’Ambrosio [46] as a non-destructive ratio suitable for investigating chlorophyll content. In addition, Buschmann [47] confirmed that the ratio of 690 nm and 735 nm induced by the loss of photosynthetic function caused by stress in the chlorophyll content was expressed as a good indicator of the fluorescence content of the leaf tissue. Agati et al. [48] has argued that he can induce a change in physiological photosynthetic activity of leaf temperature stress at 685 nm and 730 nm, as shown in Equation F685/F730. In this fluorescence ratio study, the ratio of the wavelength range with the initial max value of the spectrum measured in the fluorescence-induced reaction and the wavelength at the next inflection point was confirmed. In this study, the maximum value identified in the UV LED of garlic and the inflection point wavelength are used for comparative analysis.

### 2.5. Chrolophyll Fluorescence Spectral Pre-Processing 

The chlorophyll fluorescence spectral image of the corrected garlic was extracted using the region of interest (ROI) for each vegetable exposed to environmental stress. 400 spectral data were extracted from garlic samples grown under five environmental conditions (control group, high temperature level 1, high temperature level 2, soil moisture stress level 1, and soil moisture stress level 2) and used for the spectral analysis process. The extracted spectral matrix can be affected by various factors, including light intensity, sensor sensitivity, and changes in ambient temperature and soil moisture, which affect the acquisition of quantitative information. This process can cause mutations by obtaining incorrect information about the crop and its biological properties. Therefore, we adopted three spectral preprocessing, such as spectral smoothing, multiplicative scatter correction (MSC), and standard normal variate (SNV), to solve these problems. In this paper, three methods were used for spectral pre-processing, including spectral smoothing, multiplication scatter point correction (MSC) [49], and standard normalization (SNV) [50]. 

### 2.6. Evaluation of Stressed Garlic Classification Modeling Methods 

In this study, to classify abiotic stress in garlic, we used a Partial Lest Squares Discriminant Analysis (PLS-DA) analysis from the spectrum, pretreated with Smoothing, MSC, and SNV in the initial spectrum of garlic leaf samples observed under UV LED light sources. PLS-DA specializes in categorical classification problems [51]. It is used to predict response variables (output variables) and is a multi-linear regression technique used to model linear relationships between predictors (input variables) and response variables [52,53]. As a result of modeling between predictors and response variables, we find the principal component to maximize the difference between the physiological and chemical variables of garlic in each class (normal and stressed states) and derive the result of identifying and selecting whether the corresponding garlic belongs to the normal group or the stressed group. It constructs the model by simultaneously considering the vector explaining the variance of the predictors and the vector that maximizes the difference between classes. Spectrum data comprises high-level information, which is dimension-reduced through PLS-DA and expresses the data in low dimensions by selecting variables and finding direction vectors that explain the variance of predictors to reduce the complexity of the model. Through this, the performance of the model is improved, and the variables necessary for classification are selected. In this study, spectral variables optimized for classification are selected from the spectrum of garlic grown in each environmental group. Before constructing the PLS-DA model, the spectra were evaluated by separating them into a correction set (75%) and a prediction set (25%), with reference to the following guidelines [54,55] for 400 pieces of each garlic leaf sample from the control and stressed groups. Model optimization, construction, and classification analysis were performed using a program developed using MATLAB R2021a (The MathWorks Inc., Natick, MA, USA). 

### 2.7. Statistical Analysis

MATLAB R2021a (The MathWorks Inc., Natick, MA, USA) and OriginLab 2023 SR1 were used to analyze the PLS-DA images from the spectrum preprocessing and classification models. Using one-way analysis of variance, pixel values were extracted from the images by applying the PLS-DA classification model of garlic grown for each temperature stress to confirm whether there was a statistically significant difference in the images between each temperature variable. Statistical significance was considered at *p* < 0.05.

## 3. Results

### 3.1. Fluorescnece Spectral Features

Figure 4 displays the all raw data (a) of garlic treated with high-temperature and moisture stress in soil during the fragmentation of garlic, along with the spectrum (b) pretreated with SNV. In general, our leaf spectra signature has a similar pattern to the study of Lang et al. [56]. We discovered that a significant peak was found at around 673 nm. Furthermore, we were able to discover a shoulder at 717 nm. The waveband at 673 nm was associated with photosystem I (PSI), whereas 700 to 750 nm corresponded to the combination of photosystem I and II (PSII), as noted by Pedrós, et al. [57]. More closely, we can distinguish that a healthy garlic plant (black line) has higher intensity than the stressed groups. Due to the abnormal condition, plant tissue was likely to produce heat, which strongly influenced fluorescence emission [58]. Nonetheless, it is challenging to describe the different intensities for each stress treatment. Perhaps this is due to the limitation of chlorophyll fluorescence to investigate specific stressors in more detail [47]. Litenthaler et al. [59] suggested that the change in the shoulder wavelength, which is the maximum value of the fluorescence intensity of the fluorescence emission spectrum and the maximum value of the low wavelength, is related to the fluorescence yield according to the chlorophyll content. As shown in Section 2.4, in this study, the maximum wavelength of 673 nm and the shoulder wavelength of 717 nm were used to calculate the fluorescence ratio. More specifically, we can observe that a healthy garlic plant (black line) exhibits higher intensity compared to the stressed groups, confirming the distinction between the spectra of plants in the control group and those subjected to high-temperature and soil moisture stress. This disparity indicates that 405 nm UV LEDs can effectively capture the characteristics arising in the functional groups of chlorophyll cell components in crops. Moreover, this may affect the classification model, which will be discussed in the following paragraphs.

### 3.2. Chrolophyll Fluorescence Ratio Mode

Figure 5 shows the results of the masked fluorescence ratio image from the raw single-band image obtained from the multispectral camera from the high-temperature and moisture stress conducted for seven days. In the case of F690/F735 (a)–(e), the control group (a) showed an image close to light green as a whole. Excluding this, in the remaining environmental stress group, there were parts showing values similar to those of the control group in some areas, but no subtle difference was visually detected for each environmental stress. It is confirmed that the nature of the fluorescence wavelengths of F690 and F735 makes it difficult to distinguish the environmental stress of the crop. In the case of F685/F730 (f)–(j), it can be seen that the fluorescence ratio is evenly distributed as a whole rather than F690/F735. The change in the image occurred from the center to the end of the leaf, and the phenomenon was found in (g)–(j), which were stressed compared to the control group. This seems to be the result of the withering of garlic leaves caused by stress due to fluorescence, and F685/F730 seems to have identified this part. It is confirmed that the analysis of finer environmental stresses is difficult. F673/F717 derived from the fluorescence spectrum identified in this study was classified in compliance with the leaf image compared to the top two wavelength ratios. The control group (k) showed a reaction in a region that was not found in other fluorescence ratios, which seems to have resulted from stress caused by other factors, even though the crop was exposed to a relatively stable environment. On the other hand, in (l)–(o), where stress was applied, it can be seen that there are more red areas than in the top two fluorescence ratio images, indicating that the ratio of the fluorescence wavelength separated each area of the crop well.

### 3.3. Analysis of Chlorophyll Fluorescence Ratio Model of Garlic Phenotype

Statistical box values and statistical tables extracted from images applied with the fluorescence ratio from the images masked at the representative wavelength of 673 nm in the spectrum extracted during the high-temperature and soil moisture stress period (seven days) of garlic are shown in Figure 6 and Table 2. The fluorescence ratio statistics of garlic were extracted from the Figure 5 image data for the Cg, HL-1, HL-2, MHL-1, MHL-2. Analysis of variance, performed from the developed fluorescence ratio image pixel values, was calculated by removing some data containing noise and then extracting boundary values. In the case of the model applied with F690/F735, there was a large mean difference between the Cg and the MHL-2, as shown in (a). Although the ratio of the corresponding wavelength was shown to be comparable between normal cultivated crops and those subjected to extreme stress, there was a slight difference among the HL-2 and MHL-1 and MHL-2. In F685/F730, the results were similar to those of F690/F735, and when HL-2, MHL-1, and MHL-2 were compared to F685/F730. F673/F717 did not identify a clear difference between the Cg and the MHL-2 system under stress compared to other fluorescence ratios. 

### 3.4. Developed PLS-DA Model

Figure 7 shows the beta coefficients for a PLS-DA model developed using fluorescence spectra over the entire wavelength. These coefficients were used to identify the major wavelengths that had a significant influence on the performance of the model. Four wavelengths (693.3, 744.3, 770.1, 782.2) were selected and plotted. In Table 3, PLS-DA models were constructed from UV chlorophyll fluorescence spectra combined with Smoothing, SNV, and MSC, and this model uses 400 data variables from the environmental variable spectrum. Regarding Smoothing, the classification performance was over 70% for Cg and HL-1 MHL-1 MHL-2, which was the most common model but also yields a fairly good level of classification results. Under HL, which is high-temperature stress, HL-1 and HL-2 showed weak classification performance of 45%, which means that this model does not confirm subtle differences between the sample and the chemical composition of photosynthesis due to high-temperature stress in the crop, and it was found that spectrum classification based on temperature stress has very low potential in MSC and SNV models as well as Smoothing models. On the other hand, the HL class and MHL class classification showed a classification performance of over 80%, which means that the characteristics of the crop due to moisture stress show higher potential within this spectrum range than high-temperature stress. On the other hand, in the MSC and SNV models, the group classification performance of the HL class and MHL class was significantly lower than in Smoothing, which is confirmed by the introduction of the detailed part of Smoothing’s data scattering effect correction. MSC and SNV showed higher group classification performance than Smoothing. In particular, in the case of SNV, the classification performance of the HL class and MHL class was improved, suggesting that SNV maintains overall data shape and corrects effects between samples in compliance.

### 3.5. Chlorophyll Fluorescence Model Images

The visualization image that was developed through PLS-DA from the fluorescent chlorophyll spectrum data, pretreated with Smoothing, MSC, and SNV, and applied, is shown in Figure 8. The image to which Smoothing was applied is relatively noisy compared to other models, and the correction of leaves by light was somewhat inappropriate. In the case of (a) for Cg, it can be seen that the response is somewhat higher than that of the stressed group, but it seems that there are overfitting parts for some areas. HL-2 stage (c) shows a relatively lower value than Cg and HL-1 stage (b), but it seems difficult to accurately judge. In the case of (d), which is the first stage of soil moisture stress, it shows a relatively compliant value with Cg, which was judged to be higher than the group that received only high-temperature stress due to the transpiration of crops due to the soil moisture stress effect, even though it was at a higher temperature than Cg. It was confirmed that under double stress of the HL-2 stage and MHL-2 stage, (e) showed a very low value compared to (a)–(d). This seems to be the result of the decrease in the activity of the chlorophyll optical system, even though the excitation phenomenon was induced through UV-LED due to the stress caused by the crop. The MSC model showed more satisfactory imaging performance than Smoothing. (f) for Cg showed a very high level of intensity compared to other models, confirming that the chlorophyll optical system was activated from the light source among garlic crops and appeared to be a result. High-temperature stress (g) and (h) show relatively lower levels of intensity compared to Cg, and it was confirmed that high-temperature stress affects the growth of garlic crops. It is judged that (i), which is a soil moisture stress group, is in a high-temperature state as a result of Smoothing, but it affects the growth of the crop due to soil moisture stress. However, this could be because the crop that had been in a dark adaptation state for 12 h was expressed from specific stress caused by sudden light stress and scattering, and further research is required after this. It was found that (j), where both high-temperature stress and soil moisture stress were applied, showed a very low value, and it was confirmed that there was a clear difference compared to the control area grown in an appropriate environment. The SNV model showed similar results to the MSC model, but it contained noise in some areas. It was confirmed that SNV indirectly removed fluctuations in spectral data to indicate some irregular fluctuations in the image, whereas MSC corrected data in consideration of the fluctuations in the intensity of light due to the scattering effects of light, thereby correcting overall irregular data fluctuations. In the case of the control tool (k), it showed a higher level than MSC, and it was confirmed that there was a part that was over-fitted by light, but it showed a higher level than the stress group. In the case of HL-1 (l), it was confirmed that the area showed relatively high values in areas not visible in Smoothing and MSC. It is confirmed that SNV showed more compliance with the volatility caused by the scattering effects by ambient light than MSC. It can be seen that the areas with significantly low light intensity were also corrected in high-temperature stress stage 2 (m), MHL-1 (n), and MHL-2 (o). It is judged that noise was relatively effectively removed compared to Smoothing because it is effective in reducing predictable fluctuations by removing scaling while maintaining the structure of the overall spectrum data, whereas Smoothing showed excellent performance in effectively increasing the discrimination power of light transmission, even in areas where high-temperature and soil moisture stress were not evenly distributed. Compared to Cg (k), HL-1 (l), and (m) MHL-1 (n), SNV showed excellent performance in increasing the discrimination power between the control group and high-temperature and soil moisture stress, but further discussion is needed on how to effectively alleviate this under the influence of lighting in some areas. Therefore, it is judged that the phenotypic classification model can be effectively mitigated by considering the characteristics of complex environmental stress by capturing the underlying relationship between spectrum and chlorophyll content through fluorescence reactions by establishing an appropriate mathematical correlation between chlorophyll photometers.

### 3.6. Statistical Results of PLS-DA Model

Statistical box figures and statistical tables extracted from Figure 8 of garlic cultivated by environment, developed through PLS-DA from spectral data extracted during the high-temperature and soil moisture stress period (seven days) of garlic, are shown in Figure 9 and Table 4. Analysis of variance performed on the developed fluorescence ratio image pixel values involved removing some data containing noise and then extracting boundary values. In the case of the model image to which Smoothing was applied, different average values were shown between (a) Cg and HL-1, which seems to be a result due to the characteristics of the Smoothing model mentioned in Section 3.2. However, there was a clear difference between Cg and MHL-2, and it was much lower than other areas. HL-2 showed a low-level average value, which seems to be a result of including some information lost from Smoothing. In the case of MHL-1, a relatively low average value was shown. In the case of the MSC model, differences in compliance with Cg and HL-1 was shown compared to Smoothing. On the other hand, similar levels of results were shown in HL-2 and MHL-1, which is judged to be a result of an overestimated value due to the fluorescence of leaves due to photoreaction. Overall, SNV showed a higher average than the previous two models, which seems to be the result of SNV’s prediction by providing detailed information about places with less influence of light through normalization.

## 4. Discussion

Plants have chlorophyll, carotenoids, proteins, and biochemical properties specified by reflectance or absorption and fluorescence. The spectral fluorescence reaction by the leaf pigment obtained from the spectroscopic camera through fluorescence induction confirmed the high intensity of the plant in the near-infrared band [60]. In the near-infrared region near 700 nm, unhealthy crops exhibit low-intensity, and the same reaction was observed for garlic leaves subjected to high-temperature stress [61]. Using the peak wavelength ratio of the chlorophyll spectrum emitted during the measurement period made it possible to confirm the state of the plant compared to previously used methods by imaging information. In various studies, peaks have been identified between about 680 nm and 760 nm, corresponding to the chlorophyll fluorescence emission of plant leaves. In the existing ratios, it was possible to quickly evaluate the state of the leaves responding to UV LED for the control and the stress group, but fine classification was not possible. The F673/F717 proposed in this study could clearly determine the classification of leaves and damaged leaves compared to garlic under high-temperature and soil moisture stress. This result is similar with previous studies [62,63], which are confirmed to be affected by color because the fluorescence ratio is based on image pixel information extracted from the spectrum, and it is judged that the threshold range and properties of the cuticle causing light to be reflected from the surface of the garlic leaf during the masking process, and variables affected by wide angle and incident light angles were affected. Therefore, it is confirmed that the fluorescence ratio method for screening the growth of high-temperature and soil moisture stress in garlic is not effective. To improve this, an approach to refining the preceding steps should be attempted. The developed model image showed that it enables improved interpretation according to some garlic individuals in the visualization of stress detection of garlic. Non-biological stresses such as high-temperature and soil moisture stress can cause nonlinear changes in the spectral characteristics of plants [64,65]. Linear models such as PLS-DA assume a linear relationship between spectral data as an independent variable and non-biological stress as the dependent variable in this study, which is the complex nonlinear relationship present in the data [66]. The learned classification model for high-temperature and soil moisture stress showed a classification performance of over 90% in the second stage of control and moisture stress. In various pretreatment approaches of the spectrum, Smoothing was inefficiently identified for garlic crops, as shown in Figure 8, and further evaluation for other crops is suggested, considering the area and height of the leaf. MSC and SNV controls showed high sensitivity to the fluorescence reaction compared to the stressed group [67,68,69]. In addition, garlic under the first stage of moisture stress showed a relatively compliant fluorescence reaction visualization, although not as much as the control, and a relatively compliant pattern was observed despite being under stress. High-temperature two-stage stress showed a high-response pattern in some areas, which is judged to have an effect on this even though a port with high reflectivity in surrounding environmental elements was captured by the camera sensor and removed by masking at the time during garlic leaf photography. Previous researchers have used fluorescence reaction technology to measure the photometer activity of a leaf in a plant by using differences in light intensity and temperature. Previous studies have measured the activity of specific parts of the plant [70]. In contrast, this study resolves these differences by grasping the interaction and influence between multiple variables using spectral data of the fluorescence response. Although fluorescence imaging techniques have access to instruments that directly measure chlorophyll content, such as spectrophotometers, PLS-DA can be used as an alternative for the classification of abiotic stress visualization in crops under some high-temperature stresses. 

Table 5 presents the results of evaluating the model classification performance developed in this study for evaluating the high-temperature and soil moisture stress of garlic by measuring with UV LED. When using the chlorophyll ratio, it was possible to compare the results of the analysis of variance, as in the above results, but the relationship between the data is very low for practical use. In other words, in general, when the ratio of the spectrum is used, it is suggested that photography conditions and various variables should be removed, and in order to improve the performance of the model, research using other light sources or other crops is needed [71,72,73]. On the other hand, models using PLS-DA show higher performance than the chlorophyll ratio, but very low performance in Smoothing. This is due to the occurrence of noise cancellation while reducing the minute differences between spectral measurements. In terms of physiology, these minute differences also represent important characteristics of plants. The minute changes in chlorophyll concentration at certain wavelengths can reflect a plant’s ability to photosynthesize, its stress response, and so on. When these minute changes are reduced during the Smoothing process, the precision that reflects the characteristics of the plant decreases. Therefore, the Smoothing process is necessary for noise cancellation and data simplification, but its performance seems to have been degraded due to the loss of information generated during this process. MSC outperforms previous models, but due to the nature of MSC analysis, when UV LED light is transmitted and reflected on garlic leaves in the process of correcting the scattering effect of light, the scattering effect that appears, depending on the surface condition, particle size, and density, adds noise to the signal and affects the model results. After that, it is suggested that the reflection of garlic leaves should adjust to environmental conditions such as the intensity of the LED, which is an appropriate correction parameter, and moisture that may occur from the surface of the leaf [74]. SNV is a method of normalizing the mean and standard deviation of the spectrum generated from the fluorescence reaction to correct optical and structural differences in garlic leaves, reflecting each environmental condition and reducing the variability of measured values. It shows the highest performance among the models used in this study, but there is a limit to its application in the plant field [75]. It is possible that a plant stress resistance strategy exists during periods of high-temperature stress delivery, and many experimental studies have conducted relatively short-term tests on stress. As a result, in order to increase our understanding and the performance of identifying environmental stress interactions in garlic grown in warm regions, it is necessary to additionally perform heat-resistant and vulnerable mechanisms [76]. In order to improve the accuracy of the model, the generalization ability of the model should be verified by increasing the number of samples and considering adjustments to image acquisition parameters. It is required to improve performance through data characteristics’ or data outliers’ processing [77]. Although this approach does not visualize the photometer and the chemical and biological content within absolute chlorophyll, it presents its potential in monitoring applicable and non-destructive crops in a variety of environments [78]. However, this method has the disadvantage that it cannot operate in places without interference of light and in large areas. To improve this, preliminary studies that can clearly identify the characteristics of chlorophyll are needed. 

## 5. Conclusions

This study constructed a crop phenotyping monitoring method based on chlorophyll fluorescence and multispectral imaging to mitigate abiotic stress in garlic plants. The plants were divided into five groups and treated with a combination of control, high-temperature stress, and soil moisture stress for seven days. Subsequently, we acquired images of the fluorescence emission of dark-adapted garlic leaves over 12 h using a 405 nm wavelength LED light as the light source to induce the plants’ chlorophyll fluorescence emission. We quantified the characteristics of chlorophyll fluorescence and evaluated the stress classification performance through the ratio of chlorophyll fluorescence and the PLS-DA model. The evaluation results of the classification performance of F690/F735, F685/F730, and F673/F717 all showed very low performance, below R^2^ = 0.3, and the proposed model had difficulty detecting photosynthetic activity. Future research will supplement the light source and lighting detection system using monochromatic LED lighting in the blue band or a long-pass filter to induce chlorophyll, considering the photochemical damage to the crops. In the PLS-DA model based on fluorescence imaging, SNV showed a performance of R^2^ = 0.7, proving the ability to detect high-temperature and soil moisture stress in some garlic plants. However, it seems difficult to apply it in detecting detailed abiotic stress. Since the characteristics of chlorophyll fluorescence were found to be affected by long-term stress, it appears that ‘Namdo’ garlic, which is grown in warm regions, has adapted to high-temperature stress, and characteristic expression was delayed during the 7-day period, thus protecting the photochemistry of the garlic leaves. Future considerations include improving the additional performance of the fluorescence activation system and increasing the accuracy and speed of data processing.

## Figures and Tables

**Figure 1 sensors-24-01442-f001:**
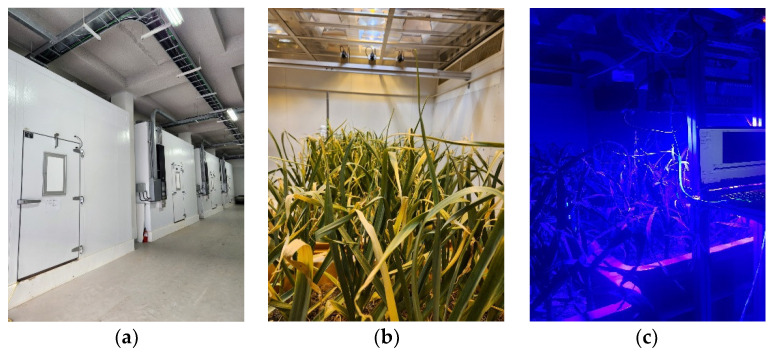
(**a**) Cultivation of plants under lighting, temperature, and soil moisture in a controlled environmental chamber; (**b**) garlic grown in the cultivation chamber; (**c**) chlorophyll fluorescence excitation system.

**Figure 2 sensors-24-01442-f002:**
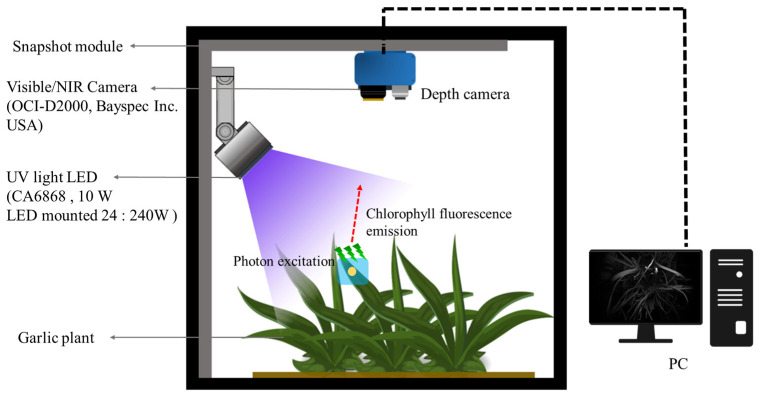
Modules and schematic diagrams for which fluorescence-induced snapshots of crops are available in the field.

**Figure 3 sensors-24-01442-f003:**
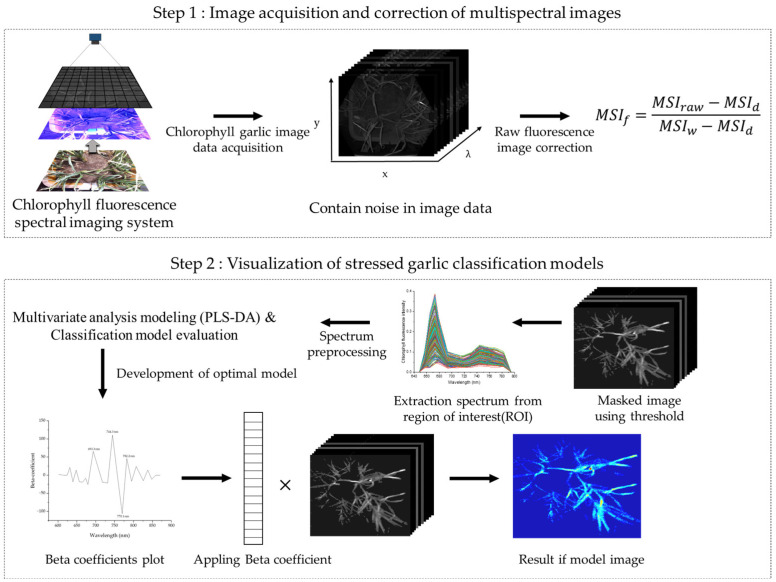
A schematic flowchart for multi-spectral fluorescence image acquisition and image classification.

**Figure 4 sensors-24-01442-f004:**
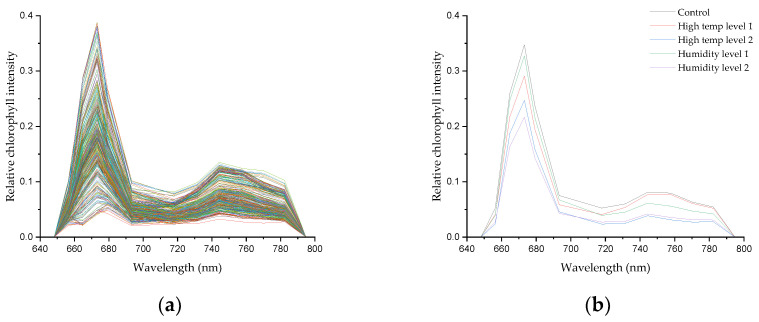
(**a**) Raw spectral data; (**b**) SNV-spectral preprocessing average for each garlic group.

**Figure 5 sensors-24-01442-f005:**
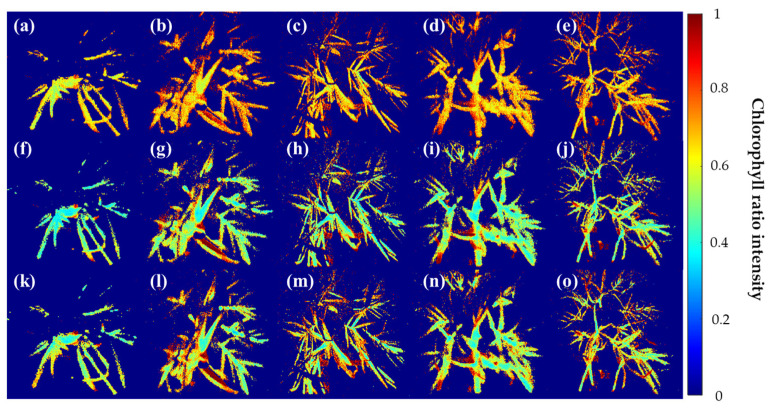
Visualization of garlic samples by environmental composition using the Chlorophyll ratio model. (**a**–**e**) F690/F735, (**f**–**j**) F685/F730, and (**k**–**o**) F673/F717 models. In all figures, Cg, HL-1, HL-2, WHL-1, WHL-2 are shown from left to right.

**Figure 6 sensors-24-01442-f006:**
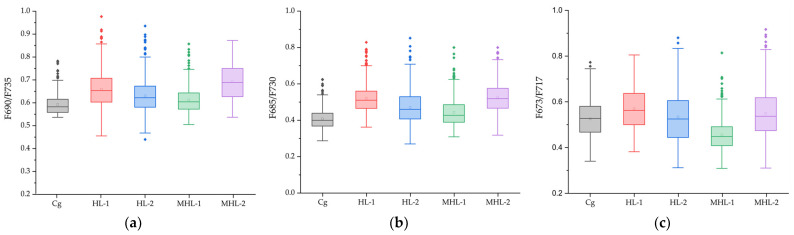
Statistical box plot of five environmental variables of garlic applied with the Chlorophyll ratios F690/F735 (**a**), F685/F730 (**b**), F673/F717 (**c**). The square boxes represent interquartile ranges (IQR), the small squares represent the mean values, the horizontal lines represent the median values, and whiskers represent values 1.5 times the IQR range. Round dots represent outliers.

**Figure 7 sensors-24-01442-f007:**
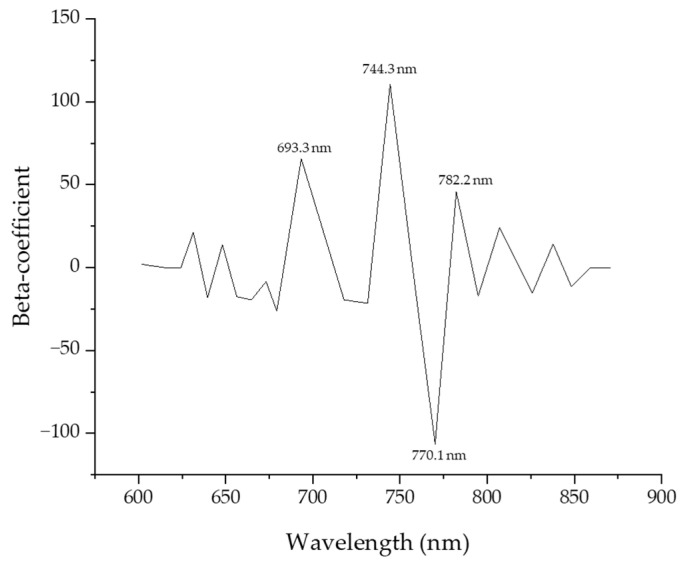
Beta coefficient curve of the Partial Least Squares Discriminant Analysis model for predicting the fluorescence response of garlic.

**Figure 8 sensors-24-01442-f008:**
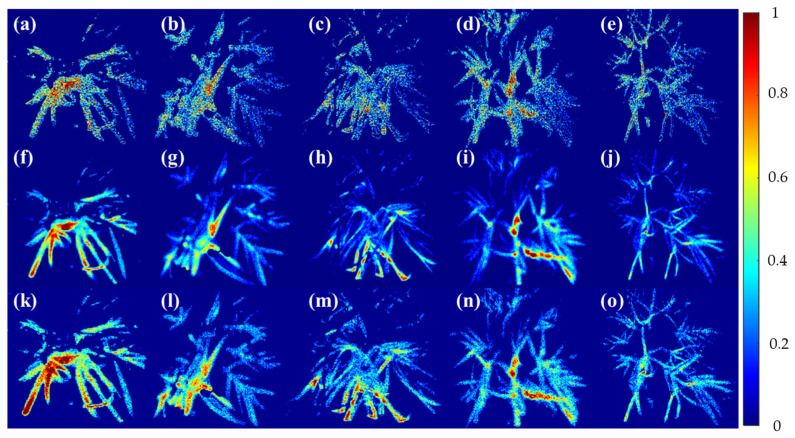
Visualization of garlic samples by environmental composition using the PLS-DA model. (**a**–**e**) Smoothing, (**f**–**j**) MSC, and (**k**–**o**) SNV models. Cg, HL-, HL-2, MHL-1, MHL-2 from left to right.

**Figure 9 sensors-24-01442-f009:**
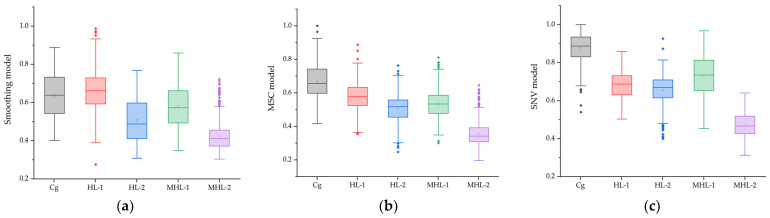
Statistical box plot of five environmental variables of garlic applied from the PLS-DA model: Smoothing (**a**), MSC (**b**), and SNV (**c**). The square boxes represent interquartile ranges (IQR), the small squares represent mean values, the horizontal lines represent median values, and the whiskers represent values 1.5 times the IQR range. Round dots represent outliers.

**Table 1 sensors-24-01442-t001:** Namdo garlic environmental stress conditions.

	Group	Chamber Environmental Composition
Garlic environmental stress group	Cg	20 °C/8 °C, 0 kPa
	HL-1	28 °C/16 °C, 0 kPa
	HL-2	36 °C/24 °C, 0 kPa
	MHL-1	28 °C/16 °C, 30 kPa
	MHL-2	36 °C/24 °C, 30 kPa

Cg: Control group, HL-1: Heat level 1, HL-2: Heat level 2, MHL-1: Moisture–Heat level 1, MHL-2: Moisture–Heat level 2. Chamber environmental composition: Temperature setting(day/night), Soil water potential.

**Table 2 sensors-24-01442-t002:** Statistics of garlic by five environmental variables applied with the Chlorophyll ratio model.

Model Group	F690/F735 Intensity			F685/F730 Intensity			F673/F717 Intensity		
	Mean ± SD	Range	*p*-Value	Mean ± SD	Range	*p*-Value	Mean ± SD	Range	*p*-Value
Cg	0.59 ± 0.04 ^a^	0.53–0.78	<0.05	0.40 ± 0.05 ^c^	0.28–0.62	<0.05	0.52 ± 0.08	0.34–0.77	<0.05
HL-1	0.65 ± 0.07	0.45–0.97		0.52 ± 0.07	0.36–0.82		0.57 ± 0.08	0.38–0.80	
HL-2	0.63 ± 0.07	0.44–0.93		0.47 ± 0.08	0.26–0.85		0.53 ± 0.11	0.31–0.88	
MHL-1	0.61 ± 0.05	0.50–0.83		0.44 ± 0.07	0.30–0.80		0.45 ± 0.06 ^e^	0.30–0.81	
MHL-2	0.70 ± 0.09 ^b^	0.53–0.86		0.52 ± 0.08 ^d^	0.31–0.80		0.54 ± 0.10 ^f^	0.31–0.91	

The values of pixel intensities in the fluorescence ratio model images for each garlic exposed to five environmental stresses were analyzed by one-way analysis of variance (ANOVA) followed by a post-test. Statistical significance was set at *p* < 0.05. Mean: mean value, SD: Standard deviation, Range: values from max to min. Model F690/F735 ^a,b^, F685/F730 ^c,d^ and F673/F717 ^e,f^. In the 0.05 Tukey honestly significant difference (HSD) test, cg and hdl2 showed significantly different values.

**Table 3 sensors-24-01442-t003:** Results of PLS-DA preprocessing models for the classified accuracy of chlorophyll fluorescence stressed garlic (%).

Models	Class	Calibration					Class	Prediction				
Smoothing		Cg	HL-1	HL-2	MHL-1	MHL-2		Cg	HL-1	HL-2	MHL-1	MHL-2
	Cg						Cg					
	HL-1	70.4					HL-1	70.1				
	HL-2	72.1	45.6				HL-2	72.9	45.1			
	MHL-1	88.6	83.3	90.7			MHL-1	86.6	80.6	85.7		
	MHL-2	94.4	88.0	82.2	41.8		MHL-2	91.8	88.1	82.0	40.7	
MSC		Cg	HL-1	HL-2	MHL-1	MHL-2		Cg	HL-1	HL-2	MHL-1	MHL-2
	Cg						Cg					
	HL-1	65.9					HL-1	62.7				
	HL-2	72.0	34.2				HL-2	73.7	32.3			
	MHL-1	87.4	62.8	80.7			MHL-1	85.8	62.7	78.2		
	MHL-2	96.8	76.3	62.1	62.4		MHL-2	93.3	81.3	63.9	60.7	
SNV		Cg	HL-1	HL-2	MHL-1	MHL-2		Cg	HL-1	HL-2	MHL-1	MHL-2
	Cg						Cg					
	HL-1	79.1					HL-1	79.9				
	HL-2	71.7	52.2				HL-2	73.7	49.8			
	MHL-1	91.0	64.9	85.0			MHL-1	89.6	67.2	85.7		
	MHL-2	97.0	85.0	66.3	68.2		MHL-2	95.4	85.1	65.4	67.8	

**Table 4 sensors-24-01442-t004:** Statistics of garlic of five environmental variables applied with the PLS-DA model.

Model Group	Smoothing Intensity			MSC Intensity			SNV Intensity		
	Mean ± SD	Range	*p*-Value	Mean ± SD	Range	*p*-Value	Mean ± SD	Range	*p*-Value
Cg	0.63 ± 0.12 ^a^	0.40–0.88	<0.05	0.66 ± 0.11 ^d^	0.41–0.96	<0.05	0.87 ± 0.08 ^i^	0.53–1.0	<0.05
HL-1	0.66 ± 0.10	0.27–0.98		0.57 ± 0.08 ^e^	0.35–0.88		0.68 ± 0.07 ^j^	0.50–0.85	
HL-2	0.50 ± 0.11 ^b^	0.30–0.76		0.50 ± 0.08 ^f^	0.24–0.76		0.65 ± 0.09 ^k^	0.39–0.92	
MHL-1	0.57 ± 0.11	0.34–0.85		0.53 ± 0.08 ^g^	0.30–0.81		0.73 ± 0.10 ^l^	0.45–0.96	
MHL-2	0.42 ± 0.07 ^c^	0.30–0.71		0.35 ± 0.07 ^h^	0.19–0.64		0.47 ± 0.06 ^m^	0.31–0.63	

The values of pixel intensities in the fluorescence ratio model images in each garlic exposed to five environmental stresses were analyzed by one-way analysis of variance (ANOVA), followed by a post-test. Statistical significance was set to *p* < 0.05. Mean: mean value, SD: standard deviation, Range: values from max to min. Model smoothing ^a,b,a,c^, MSC is except ^e,g,f,g^, SNV is except ^j,k,j,l^, and they failed to classify ^d,h,i,m^. In the 0.05 Tukey honestly significant difference (HSD) test, cg and hdl2 showed significantly different values.

**Table 5 sensors-24-01442-t005:** Comparison and evaluation of models developed with UV spectra.

Model	Chlorophyll Ratio		Model	PLS-DA	
	*R* ^2^	RMSE		*R* ^2^	RMSE
F690/F735	0.209	0.523	Smoothing	0.396	0.208
F685/F730	0.256	0.577	MSV	0.562	0.168
F673/F717	0.149	0.492	SNV	0.707	0.133

## Data Availability

Data are contained within the article.

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
