# Peer review of "Chlorophyll Fluorescence Imaging for Environmental Stress Diagnosis in Crops"

_sensors, 2024, doi:10.3390/s24051442_

Round 1
Reviewer 1 Report
Comments and Suggestions for Authors
Dear editors,
I suggest the authors make major revisions, otherwise it cannot be published. My specific suggestions are as follows:
1. I suggest changing the title "Chlorophyll fluorescence imaging of environmental stress-crop diagnosis classification" to "Chlorophyll fluorescence imaging for environmental stress diagnosis in Crops".
2. In the abstract section, 408 nm LED was used to induce fluorescence, but it was not explained why this specific wavelength was chosen.
3. In the abstract section, specific experimental data results should be reflected.
4. I suggest providing a more detailed introduction to the importance of plant phenotypes and the application background of fluorescence imaging technology in plant health monitoring.
5. Section 2.1 Garlic cultivation: Please provide more information about the conditions for garlic cultivation, such as soil type, irrigation frequency, and so on.
6. Section 2.2 Chrolophyll fluorescense spectral imaging system: Please provide more details, such as the specific configuration and imaging parameters of the fluorescence module.
7. Section 2.3 Fluorescence image data correction: Explain the specifications and calibration methods of the PTFE board used in the calibration process.
8. Please increase the text size in the Figures 2 , 3...... to improve clarity.
9. 3.1. Fluorescnece spelctral features: explain more detailed spectral data, how to extract key fluorescence features from the spectrum?
10. Add a legend in Figure 5 to illustrate the fluorescence intensity range represented by different colors.
11. Add legends to the statistical box plots in Figure 6 and Table 2
12. In the discussion section, it is necessary to conduct a thorough analysis of the performance differences of the PLS-DA model under different environmental pressures, as well as the physiological mechanisms behind these differences.
13. The units of Mean, SD, Range? Such as Table 4
14. The experimental results are poor, with a maximum of 0.707. How to improve the model to improve classification accuracy? Suggest in-depth discussion
15. In the conclusion section, the main findings of the study were not clearly summarized, and no clear future research directions were proposed.
16. Please check the quality of all charts to ensure sufficient resolution.
17. The innovation of the paper is insufficient, especially in the application of fluorescence imaging technology in garlic environmental pressure diagnosis.
18. This paper failed to fully explore the differences in classification performance of PLS-DA models under different environmental pressures, as well as the physiological and ecological mechanisms behind these differences.
19. There are some unclear language expressions, grammar errors, and formatting issues.
Comments on the Quality of English LanguageThere are some unclear language expressions, grammar errors, and formatting issues.
Author Response
We sincerely appreciate your review of the paper. We have successfully identified the underlying issue and resolved it.
"Please see the attachment."

Reviewer 2 Report
Comments and Suggestions for Authors
I carefully and with great pleasure read the manuscript of Beomjin Park et al. entitled "Chlorophyll fluorescence imaging of environmental stressed crop diagnosis classification", in which the work was aimed at developing a method of garlic plant condition diagnosis using fluorescence imaging.
In my opinion, the work is performed at a sufficiently high scientific and technical level and written in an accessible language. However, from the reader's perception point of view, the paper has some shortcomings that should be corrected, before publication in Sensors journal.
Major remarks:
1) In my opinion, the introduction is written rather rambling and unnecessarily long. I think it should be structured more clearly by paying attention to the different types of stress in plants, perhaps the changes that occur within plants when they are stressed, and the methods used to determine the level of stress. When citing literature, pay attention and cite quantitative results where appropriate.
2) In the results (Figures 6 and 9 as well as tables), statistically significant differences between groups should be labeled somehow where they are observed, otherwise it seems as if no statistically significant differences were observed anywhere.
3) The fluorescence spectra shown in Figure 4 can be smoothed, then you can more accurately talk about the change in the position of the local minimum at 720 nm. In addition, it is unclear why the study refers to the local minimum as an inflection point?
Minor comments:
1) Line 52: not atoms, but chlorophyll molecules, etc.
2) Lines 58-60: incorrect wording. Visible and NIR spectroscopy are two different bands. It is better to say that NIR spectroscopy complements visible spectroscopy and allows more parameters to be analyzed.
3) Line 72: PLS - need to give a transcript.
4) Line 365 - it says 120 minutes, but Figure 1 says 121 minutes.
5) Table 1: humidity and temperature regimes are better to specify for all groups.
6) Figure 2 UV light LED - no model name in the text.
7) Figure 3 - picture quality needs improvement.
8) Figure 5 - study groups should be signed for better perception.
9) Difference in terminology: somewhere "groups" and somewhere "classes".
10) References to literature do not follow the format of the journal.
I believe that if these shortcomings are corrected, the article can be published in the journal Sensors.
Author Response

(The authors gave the same response as above.)

Round 2
Reviewer 1 Report
Comments and Suggestions for Authors
Dear editors,
The authors carefully addressed all my comments and the manuscript currently meets my requirements in terms of organizational structure and content.
But the model results have not been improved, and I noticed that the authors have indicated the need for further research. The authors can try using machine learning algorithms such as SVM, RF, or Deep Learning to improve the prediction accuracy of the model. I expect the authors' next manuscript for further research to be published in our journal, in order to make the study more continuous and valuable.
Reviewer 2 Report
Comments and Suggestions for Authors
Thank you, all comments have been taken into account